# Gene Expression in Parthenogenic Maize Proembryos

**DOI:** 10.3390/plants10050964

**Published:** 2021-05-12

**Authors:** Irina Volokhina, Yury Gusev, Yelizaveta Moiseeva, Olga Gutorova, Vladimir Fadeev, Mikhail Chumakov

**Affiliations:** 1Institute of Biochemistry and Physiology of Plants and Microorganisms, Russian Academy of Sciences, 13 Prospekt Entuziastov, 410049 Saratov, Russia; volokhina_i@ibppm.ru (I.V.); gusev_yu@ibppm.ru (Y.G.); em-moiseeva@mail.ru (Y.M.); vvf2593@gmail.com (V.F.); 2Genetics Department, Saratov State University, 83 Ulitsa Astrakhanskaya, 410012 Saratov, Russia; olga.gutorova@mail.ru

**Keywords:** *Zea mays*, gynogenesis, DNA methylation, chromatin-modifying enzymes, fertilization-independent embryo and endosperm development

## Abstract

Angiosperm plants reproduce both sexually and asexually (by apomixis). In apomictic plants, the embryo and endosperm develop without fertilization. Modern maize seems to have a broken apomixis-triggering mechanism, which still works in *Tripsacum* and in *Tripsacum*–maize hybrids. For the first time, maize lines characterized by pronounced and inheritable high-frequency maternal parthenogenesis were generated 40 years ago, but there are no data on gene expression in parthenogenic maize proembryos. Here we examined for the first time gene expression in parthenogenic proembryos isolated from unpollinated embryo sacs (ESs) of a parthenogenic maize line (AT-4). The DNA-methylation genes (*dmt103*, *dmt105*) and the genes coding for the chromatin-modifying enzymes (*chr106*, *hdt104*, *hon101*) were expressed much higher in parthenogenic proembryos than in unpollinated ESs. The expression of the fertilization-independent endosperm (*fie1*) genes was found for the first time in parthenogenic proembryos and unpollinated ESs. In parthenogenic proembryos, the *Zm_fie2* gene was expressed up to two times higher than it was expressed in unpollinated ESs.

## 1. Introduction

Angiosperm plants reproduce both sexually and asexually (by apomixis) [1,2,3]. In sexual plants, in the absence of fertilization neither embryo nor endosperm can develop. In apomictic plants, mainly polyploid species, the embryo and endosperm develop under fertilization-free conditions (fertilization-independent seed (*fis*) and endosperm (*fie*)). Apomixis is of major interest for the fixation of genotypes obtained by using heterosis (hybrid vigor). Control of apomixis would make it possible to fix highly productive and environmentally adaptable genotypes for modern agriculture. An understanding of how apomixis acts at the molecular level could enable apomixis to be controlled.

Apomixis has been observed in the wild ancestor of maize, *Tripsacum dactyloides* L., and in *Tripsacum*–maize hybrids [4,5]. Hybrids with 38 chromosomes (2*n* = 20 maize + 18 *Tripsacum*) and with fertilization-free seed development were very interesting as a model [4,5,6,7]. Some researchers speculated that apomictic reproduction in *Tripsacum* is controlled by nine chromosomes [8]; others showed that one chromosome is responsible for apomixis-like development [9,10]. Which genes govern the switch from sexual to apomictic reproduction in *Tripsacum*–maize hybrids still remains unclear; there are data that the responsible genes are those coding for the chromatin-modifying enzymes (CMEs) [7].

It has been suggested that apomixis may result from temporal or spatial deregulation of the transcription programs that control sexual reproduction [11,12,13,14]. In particular, the CME genes are differently expressed in *Tripsacum*–maize hybrids and in sexual maize [7,15,16]. The CME genes have clear qualitative expression differences before and after pollination in maize and in *Tripsacum*–maize hybrids [7]. In contrast to sexual maize, the expression of three DNA methyltransferases (*dmt102*, *dmt105*, *dmt103*) and one chromatin-modulating factor (*chr106*) is suppressed in *Tripsacum*-maize hybrids at all developmental stages. Two genes [histone linker (*hon101*) and histone deacetylase (*hdt104*)] are expressed heterochronically [7].

The fertilization-independent development of the maize seed and endosperm is correlated with the methylation of the genes controlling endosperm development and the imprinting phenomenon [17,18,19,20]. Although gene imprinting (parents’ allele-dependent gene expression) is observed in the maize embryo and endosperm, it has not been found in adult plant tissues [21,22,23]. On the other hand, in the maize embryo, the genes are more methylated than in the endosperm [24].

The *Zm_fie* genes belong to the Рolycomb Repressive Complex 2 (PRC2) genes, which regulate the early stages in *Drosophila* and *Arabidopsis* development [25,26]. In particular, the *Zm_fie1* gene is not expressed in the maize sperm, egg, or central cell before fertilization but is expressed in the endosperm at 2 days after pollination (DAP), with maximum activity at 10–11 DAP [25]. On the contrary, the *Zm_fie2* gene is expressed in the egg and central cells before pollination and is probably repressed during maize endosperm development [25,26]. As suggested by phylogenetic analysis of the maize *fie* genes, the *fie1* and *fie2* genes arose through the duplication of one of the ancestral paralog genes during maize genome evolution [25]. 

Locus-specific imprinting in maize is observed in the genes *fie*, *meg1* (maternally expressed gene 1), and *nrp1* (no-apical-meristem-related protein 1) [25,27,28], which are expressed in the female gametophyte without fertilization.

Modern maize seems to have a broken apomixis-triggering mechanism. Forty years ago, however, Russian breeders from Saratov State University generated the first parthenogenic maize line (AT-1) with pronounced and inheritable high-frequency maternal parthenogenesis in the ESs [29]. The AT-1 line and its derivatives AT-3 [30,31] and AT-4 [32] are polyembryonic, with parthenogenic embryo and endosperm development. There are no data on gene expression in parthenogenic maize proembryos.

Previously, we examined for the first time the expression of the CME genes in the female gametophytes of the AT-3 line and have found no large differences between the ESs of the parthenogenic line AT-3 and the ordinary line DHH-1 before and after pollination, except that the expression of the *hon101* and *hdt104* genes was higher in DHH-1 [33]. However, there are no data on maize gene expression in parthenogenic proembryos or in unpollinated ESs. It is interesting to note that although the development of parthenogenic embryos in maize ESs has been described [29,30], we did not find any homologous *Arabidopsis fis* genes in the maize database (data not shown).

The maize CME genes are possibly responsible for apomictic-like egg-cell development in *Tripsacum*–maize hybrids [5,7,23]. Those findings suggest that the CMEs are implicated in the fate of the precursor cells of the maize ESs. We chose six CME loci (*dmt102*, *dmt103*, *dmt105*, *chr106*, *hdt104*, and *hon101*) to search for qualitative differences in gene expression between fertilization-free proembryos and ESs of the parthenogenic maize line AT-4. The reason was that only these genes differ qualitatively in expression between sexual maize B73 and apomictic *Tripsacum*-maize hybrid [7].

Here we observed for the first time the expression of the CME and *Zm_fie* genes in parthenogenic proembryos and ESs isolated from the unpollinated maize line AT-4. 

## 2. Results and Discussion

### 2.1. Microscopy of ESs

In total, 50 frozen and fixed AT-4 ESs were analyzed 8 days after the appearance of pistillate stigmas (DAAPS). A typical ES contained one egg, two synergides, two polar nuclei, and an antipodal complex (Figure 1A). There were deviations from the typical ES structure, and parthenogenic proembryos developed under pollination-free conditions (8 DAAPS). At 8 DAAPS, the number of ESs with 2- to 16-cell parthenogenic proembryos was up to 18%. Figure 1B shows 8-cell proembryos at the early proembryo stage.

There also were ESs with additional cells and nuclei in the egg apparatus, synergide-like cells, and ESs with two-celled proembryos or additional egg cells (data not shown). The frequency of spontaneous proembryo formation in the AT-4 line was 10 times higher than the number of parthenogenic proembryos found earlier in the parthenogenic AT-3 line [33].

Approximately within 24 h after pollination (HAP), the zygote divides transversely, yielding a small terminal cell and a large basal cell. Forty-eight HAP, there occur additional rounds of cell division in the embryo, which generally result in a two- (or three-) celled proembryo and a two- (or three-) celled suspensor [34]. We observed 2- to 16-cell parthenogenic proembryos in the AT-4 ESs at 8 DAAPS, but we do not know exactly when a particular egg cell started dividing under pollination-free conditions. Microscopy showed that the egg cells in the AT-4 ESs were at different maturity stages and differed in their readiness for spontaneous division, because we observed 2- and 16-cell proembryos at 8 DAAPS.

Within 6–12 HAP, the pollen tube delivers two sperm cells to the ES, and within 10 s, one maize sperm attaches to the egg cell membrane, after which it fuses with it [35]. The first egg-cell division takes up to 48 HAP to produce a 2-cell proembryo [36,37] and 96 HAP to produce a late proembryo [35]. At present, we do not know how the parthenogenic proembryo and the embryo developed after pollination differs in the period of development. For 350,000 samples taken from the ears of the AT-3 maize line, no parthenogenic seed development was found (Yu.V. Smolkina, personal communication). The parthenogenic embryos disintegrated at 12–20 DAAPS, because they lacked support from the undeveloped endosperm.

Thus, at 8 DAAPS, the AT-4 line showed maternal parthenogenesis (spontaneous proembryo development from an unpollinated egg cell). The question arises: what changes in gene expression can accompany the development of parthenogenic proembryos? The DNA-methylation and CME genes, whose expression varies with apomictic development of the maize embryo [7], and the *fie* genes will be analyzed in the next sections.

### 2.2. Expression of DNA-Methylation Genes in ESs

To address the possible role of chromatin structure in the occurrence of apomixis-like proembryos in the AT-4 (unpollinated) ESs, we investigated the expression of the DNA-methylation genes in parthenogenic proembryos isolated from unpollinated ESs at 10 DAAPS, in ESs at 3 DAP, and in the embryo (7 DAP) and endosperm (10 DAP) isolated from AT-4 ESs. We wanted to determine whether there were any differences in the expression of the DNA-methylation genes (i) between the ESs with parthenogenic proembryos developed from unpollinated egg cells (10 DAAPS) and the unpollinated ESs at 7 and 10 DAAPS and (ii) between the parthenogenic proembryos at 10 DAAPS and the embryos and endosperm at 7 and 10 DAP.

The absence of qPCR products from the DNA-methylation genes *dmt102*, *dmt103*, and *dmt105* in *Tripsacum*–maize ovules indicates apomictic development [7]. In our work, the expression patterns of the *dmt103* and *dmt105* genes in the parthenogenic proembryos were significantly higher than in unpollinated ESs (10 DAAPS; Figure 2A,B), except for the *dmt102* gene (data not shown). In particular, in four biological samples consisting of five ESs with parthenogenic proembryos, the *dmt103* gene was 1.6–3.3 times more expressed (average, 2.04 ± 0.65; *p ≤* 0.05) than it was expressed in unpollinated ESs (Figure 2A). Yet, *dmt103* expression in the ESs after pollination (3–10 DAP) gradually increased nonsignificantly from ESs (3 DAP) to embryos (7 DAP), becoming up to 1.5 times higher (*p ≤* 0.05) than in unpollinated ESs (10 DAAPS) in the case of the endosperm (10 DAP). Surprisingly, *dmt103* expression was higher in unpollinated ESs at 7 DAAPS (Figure 2A).

In four samples with parthenogenic proembryos, the *dmt105* gene was 1.4–3.7 times more expressed (average, 2.44 ± 0.82) than it was expressed in unpollinated ESs (Figure 2B). Yet, *dmt105* expression in the ESs after pollination (3–10 DAP) gradually increased, becoming more than two times higher than in unpollinated ESs (10 DAAPS). We speculate that the activity of *dmt105* methyltransferase in parthenogenic embryos at 10 DAAPS may be higher than in unpollinated ESs at 10 DAAPS.

Microscopy showed that the parthenogenic proembryos in the AT-4 ESs at 8 DAAPS were at different division stages (Figure 1B). We used prefrozen (−20 °C) ears to isolate intact cooled ESs with visually observed parthenogenic proembryos and analyze the expression of the DNA-methylation genes. Unfortunately, it was impossible to recognize the development stage for parthenogenic proembryos in the intact ESs before RNA extraction. As can be seen from Figure 2, the expression levels for the *dmt103* and *dmt105* genes differed significantly (*p* ≤ 0.05), especially in samples 3 and 4 (parthenogenic proembryos). Therefore, the parthenogenic proembryos (numbered 1–4) from intact ESs possibly were at different division stages as well (Figure 2).

The DNA-methylation genes and the *chr106* gene were broadly expressed during sexual development (sporogenesis; mature ESs at 3 DAP) in the B73 maize line but were totally absent during apomictic reproduction of a *Tripsacum*-maize hybrid (38C) [7]. *Dmt102* was not expressed at any of the three stages in 38C. In this work, however, gene expression was detected in mature ESs, and it remained at the same level in parthenogenic proembryos and in reproductive embryos after pollination.

The expression of the *dmt103* gene was observed in the B73 ovules with mature ESs, but at 3 DAP it dropped to the level of C38 hybrids [7]. In our work, *dmt103* expression in unpollinated ESs was no different from that in embryos from pollinated ESs at 3 DAP; however, it increased in parthenogenic proembryos. The expression of the *dmt103* gene was high in the ovules of the B73 line with mature ESs but lower in ovules with embryos at 3 DAP [7]. In our work, *dmt105* expression did not differ between unpollinated ESs (7 DAAPS) and embryos from pollinated ESs (3 DAP). In parthenogenic proembryos, the expression of this gene was greatly increased (Figure 2B).

### 2.3. CME Gene Expression in ESs

#### 2.3.1. Chromatin-Modulating Factor (chr106) Expression

In parthenogenic proembryo 1 (10 DAAPS), the *chr106* gene was 1.4 times more expressed (*p ≤ 0.05*) than it was expressed in unpollinated ESs at 10 DAAPS. *chr106* expression in sample 1 with proembryos was similar to that in embryos from pollinated ESs (7 DAP; Figure 3). We did not find any significant differences in *chr106* expression between unpollinated ESs (7 and 10 DAAPS) and ESs with parthenogenic proembryos (sample 2) (Figure 3). In pollinated ESs (3 DAP), the *chr106* gene was two times more expressed (*p* ≤ 0.05) than it was expressed in unpollinated ESs at 10 DAAPS.

#### 2.3.2. Histone Deacetylase (hdt104) and Histone Linker (hon101) Expression in Proembryos and ESs

In parthenogenic proembryos (sample 1 but not sample 2) (10 DAAPS), the *hon101* and *hdt104* genes were 1.4 and 1.7 times more expressed (*p* ≤ 0.05), respectively, than they were expressed in unpollinated ESs at 10 DAAPS (Figure 4). Similarly, the *hdt104* and *hon101* genes were up to 2 and 2.4 times more expressed, respectively, in pollinated ESs at 3 DAP than they were expressed in unpollinated ESs at 10 DAAPS (Figure 4).

The parthenogenic embryos did not differ in *chr106*, *hdt104*, or *hon101* expression in unpollinated ESs (10 DAAPS), as compared with embryos in pollinated ESs at 7 DAP in both independent experiments (Figure 3 and Figure 4).

Thus, except for the *dmt102* gene, the expression levels of the *dmt103*, *dmt105*, *chr106*, *hon101*, and *hdt104* genes were 1.4 to 3.7 times higher (*p* ≤ 0.05) in parthenogenic proembryos (10 DAAPS) than in unpollinated ESs (10 DAAPS). Of note, in the apomictic hybrid, the expression of the CME genes was repressed, as compared with that in the sexual B73 line [7]. 

The differences in the expression of the DNA-methylation (except *dmt102*) and CME genes at 10 DAAPS could clarify parthenogenic proembryo development in unpollinated AT-4 ESs (Figure 2 and Figure 3).

##### *fie1* and *fie2* Expression in ESs

The first nucleus division in the pollinated ESs began after 16–17 HAP, as soon as karyogamy in the central cell was complete [36]. Although we did not observe parthenogenic central cell development in 50 ESs isolated from AT-4 ovules, we believe that it does exist, because the AT-4 line is characterized by parthenogenic embryo and endosperm development [32]. 

We observed for the first time the expression of the *Zm_fie1* gene in unpollinated AT-4 ESs at 3–10 DAAPS (*p* ≤ 0.05). We did not find any significant (*p* ≤ 0.05) differences in *Zm_fie1* expression between unpollinated ESs (7 and 10 DAAPS) and parthenogenic proembryos (Figure 5A). Surprisingly, we observed higher-level *Zm_fie1* expression in unpollinated ESs at 3 DAAPS (Figure 5A). In the endosperm from pollinated ESs (7–10 DAP), the *Zm_fie1* gene was expressed three times more than it was expressed in unpollinated ESs at 10 DAAPS (Figure 5A). This finding is in agreement with the earlier data [25,26]. In particular, the *Zm*_*fie1* gene is expressed in the maize endosperm at 2 DAP, with maximum activity at 10–11 DAP. It is interesting to note that *Zm_fie1* expression in the embryo from pollinated ESs at 7 and 10 DAP was two times higher than in unpollinated ESs (10 DAAPS) (Figure 5A).

We observed for the first time increased *Zm_fie2* expression in parthenogenic proembryos, especially in samples 3 and 4 (Figure 5B). In these samples, *fie2* expression at 10 DAAPS increased by 1.8 and 1.4 times, respectively (*p* ≤ 0.05), as compared with unpollinated ESs (10 DAAPS; Figure 5B). As expected, the *Zm*_*fie2* gene was more expressed in the endosperm and embryo in 10 DAP (Figure 5B).

## 3. Materials and Methods

### 3.1. Plant Material

The diploid maize (*Zea mays* L.) line АТ-4 was produced by pollinating the tetraploid line KrP-1 with the diploid parthenogenic line AT-1. In the hybrid’s offspring, diploid plants (dihaploids) emerged, which were selected and self-pollinated [32]. The AT-1 line is the product of crossing the Stock 6 line [38] with the Brown Tester line [29]. The purple color of the AT-1 grains, leaves, and stems is determined by the *A*, *B*, *PL*, and *R* genes [38].

Seeds were planted by using tractor-mounted seeders (SKS-6-10, Russia) on May 14, 2020, in three 4 × 5 m plots. The planting density was 5–10 plants/m^2^. The AT-4 line was grown in the fields of the Russian Research, Design, and Technology Institute of Sorghum and Maize (Rossorgo Institute; Saratov, Russia) without watering or herbicide treatment. Weeds were removed on June 5, first by tractor and then by hand. 

The maize ears were isolated with parchment bags before pistil filaments developed (late July–early August) and were grown without pollination for 3–10 days. The unpollinated ears were collected at 3, 7, and 10 DAAPS, placed in containers with ice, transported to the laboratory, and frozen at −20 °C. For ES and proembryo isolation, the ears were harvested at 7 and 10 DAAPS. For obtaining a sexual embryo and endosperm, preisolated pistils were pollinated manually with pollen collected from other AT-4 plants. The ears were collected at 3, 7, and 10 DAP, placed in containers with ice, transported to the laboratory, and frozen at −20 °C. ESs, parthenogenic proembryos, embryos, and endosperms were isolated from prefrozen (−20 °C) ears under an MBI-9 microscope (Russia) by using steel dissecting needles and were placed into Eppendorf tubes with a cooled RNA-isolation buffer.

### 3.2. Expression of CME, DNA-Methylation, and Fie Genes in Female Gametophyte Tissues

ESs, proembryos, embryos, and endosperms were isolated from freshly collected and prefrozen ears at 7 and 10 DAAPS and at 7 and 10 DAP. mRNA was extracted from 5 proembryos and 12 ESs (50 µg at 10 and 7 DAAPS, respectively), 5 embryos (50 µg at 7 and 10 DAP), and 2 endosperm samples (or 9 mg at 7 DAP) by using a commercial NucleoSpin RNA Plant kit (catalog no. 740949.50; Macherey Nagel, Dueren, North Rhine-Westphalia, Germany) with DNase. The total RNA, harvested from 12 proembryos (10 DAAPS) and 5 embryos (7 DAP), was 2.5 ng/µL. For cDNA synthesis, 1 µg of total RNA was subjected to reverse transcription with reverse transcriptase (catalog no. EP0442; Thermo Fisher, Waltham, MA, USA) and with an oligo (dT) primer (catalog no. AM5730G; Thermo Fisher). The cDNA concentration was measured on a Qubit fluorimeter (Thermo Fisher Scientific, Singapore) by using a Qubit DNA HS assay kit, (Q32850; Thermo Fisher Scientific, Waltham, MA, USA). For all samples, the cDNA concentration was made equal (2 ng) by diluting the high concentration. Quantitative PCR (qPCR) was done on an Applied Biosystems PCR amplifier (Applied Biosystems, Foster City, CA, USA) by using a set of reagents for real-time PCR in the presence of the SYBR green dye and the ROX reference dye (catalog no. M-435; Syntol, Russia) and specific primers (see Supplemental Data 1) handpicked by using the Primer3Plus (http://www.bioinformatics.nl/cgi-bin/primer3plus/primer3plus.cgi, accession date was 24 March 2020) and Primer-BLAST programs. The primers were chosen so that the DNA and mRNA PCR products could differ in size (see Appendix A). Target gene expression profiles were determined relative to the expression level of the GAPDH gene as an endogenous control. For each data point, qPCR was done in four technical replicates for one cDNA sample. Two to four independent biological experiments were conducted (see figure notes). The relative gene expression levels were calculated by the 2^−ΔΔCT^ method [39]. Following Weaver et al. [40], we discarded the difference <1.3 times.

### 3.3. Microscopy

The isolated ESs were stained with a glycerol–acetocarmine solution, as described by Volokhina et al. [33]. The ESs were analyzed with a DM2500 light fluorescence microscope (Leica Microsystems, Wetzlar, Heese, Germany) at the Simbioz Center for the Collective Use of Research Equipment in the Field of Physical–Chemical Biology and Nanobiotechnology (Institute of Biochemistry and Physiology of Plants and Microorganisms, Russian Academy of Sciences, Saratov).

## 4. Conclusions

Although apomixis has been observed in *Tripsacum* (the wild ancestor of maize) and in *Tripsacum*–maize hybrids [4,5], modern maize has a broken apomixis mechanism. The fertilization-independent development of the embryo and endosperm was discovered 40 years ago for the AT-1 and AT-3 maize lines [29,30] and for the recently generated AT-4 line [32]. However, there are no data on the genes controlling parthenogenic egg and central cell division. It is believed that the fertilization-independent development of the maize embryo and endosperm is correlated with the methylation of the genes controlling endosperm development and the imprinting phenomenon [17,18,19,20]. In particular, the DNA-methylation genes (*dmt102, dmt103,* and *dmt105*) and the genes coding for the CMEs (*hdt104, chr106*, and *hon101*) have clear qualitative expression differences in ESs before and after pollination between maize and *Tripsacum*–maize hybrids [7]. For this reason, we examined for the first time the expression of the DNA-methylation and CME genes in the parthenogenic proembryos of the AT-4 line. The expression of the DNA-methylation (except *dmt102 gene*) and CME genes differs between parthenogenic proembryos and unpollinated ESs and could be the reason for the development of parthenogenetic proembryos in the AT-4 line.

We observed for the first time the expression of the *Zm_fie1* gene in unpollinated AT-4 ESs with or without parthenogenic proembryos. *Zm_fie2* expression in parthenogenic proembryos was up to two times higher than in unpollinated ESs (10 DAAPS).

## Figures and Tables

**Figure 1 plants-10-00964-f001:**
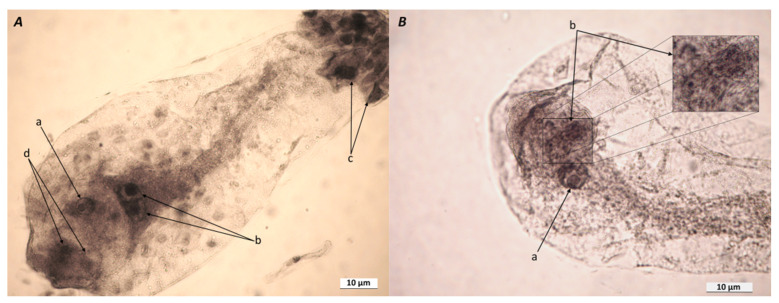
(**A**) Typical embryo sac (ES) isolated from AT-4 ovules at 8 DAAPS. Notes: a, egg cell; b, two polar nuclei of the central cell; c, antipodal complex; d, two synergides. (**B**) Unpollinated ES isolated from AT-4 ovules at 8 DAAPS. Notes: a, central cell nucleus; b, 8-cell parthenogenic proembryo and an enlarged fragment of it. The ESs (A and B) were isolated from acetic-alcohol-fixed ovules, washed, and placed in glycerol–acetocarmine (5:1, *v*/*v*) for 24 h.

**Figure 2 plants-10-00964-f002:**
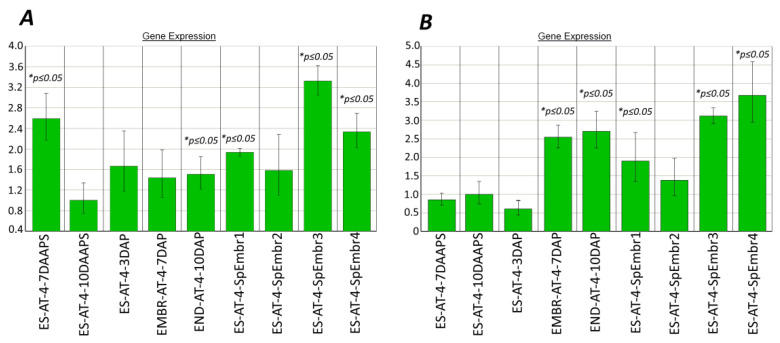
Expression of DNA-methylation genes in the parthenogenic proembryos (10 DAAPS), ESs (7 and 10 DAAPS), and embryos and endosperm (7 and 10 DAP) of the AT-4 line. (**А**) *dmt103*; (**B**) *dmt105*. Control: unpollinated ESs (10 DAAPS). * Significant differences between target tissue and control cells (ESs at 10 DAAPS) are expressed as *p ≤* 0.05. ES-AT-4-SpEmbr1-4: in one biological experiment, each of the four samples consisted of five ESs with parthenogenic proembryos. Data are presented from two independent biological experiments. ES-AT-4-7-10 DAAPS: in one biological experiment, each sample consisted of 12 unpollinated ESs. Data are presented from two independent biological experiments.

**Figure 3 plants-10-00964-f003:**
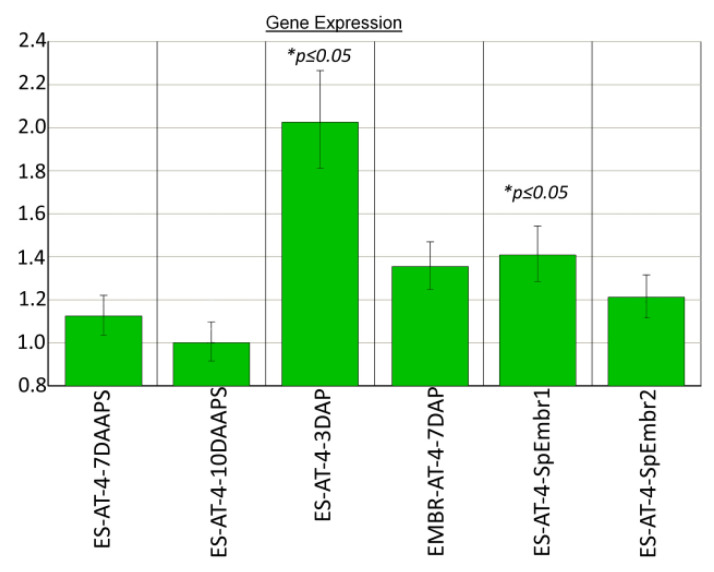
Chromatin-modulating factor (*chr106*) expression in parthenogenic proembryos (10 DAAPS) and ESs (7 and 10 DAAPS) of the AT-4 line. Control: ESs at 10 DAAPS. ES-AT-4-7-10DAAPS: in one independent biological experiment, each sample consisted of 12 unpollinated ESs. Data are from four independent biological experiments.

**Figure 4 plants-10-00964-f004:**
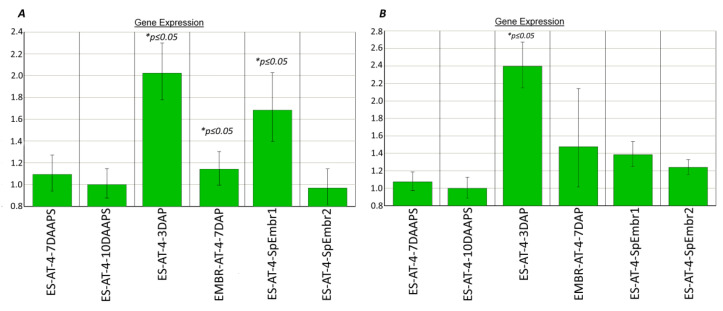
Histone deacetylase (*hdt104*) (**A**) and histone linker (*hon101*) (**B**) expression in parthenogenic proembryos* (10 DAAPS) and ESs (7–10 DAAPS). Controls: ESs at 10 DAAPS for (**A**) and (**B**). *ES-AT-4, 7 and 10 DAAPS: in one biological experiment, each sample consisted of 12 unpollinated ESs. Data are from four independent biological experiments.

**Figure 5 plants-10-00964-f005:**
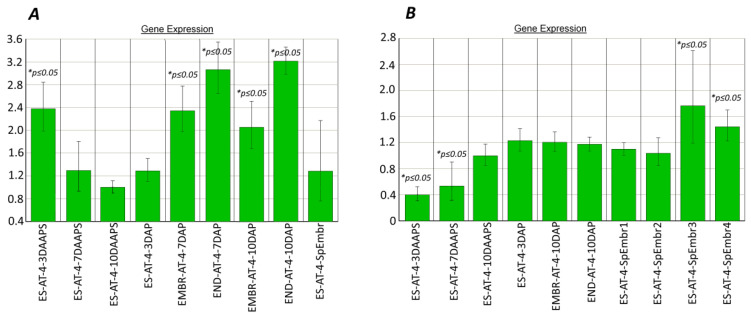
*fie1* (**A**) and *fie2* (**B**) expression in ESs with parthenogenic proembryos* (10 DAAPS) and in unpollinated ESs (7 and 10 DAAPS). Control: ESs at 10 DAAPS. *ES-AT-4-7-10DAAPS: in one biological experiment, each sample consisted of 12 unpollinated ESs. Data are from two independent biological experiments.

## Data Availability

The data presented in this study are available on request from the corresponding author.

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
