# Peer review of "Gene Expression in Parthenogenic Maize Proembryos"

_plants, 2021, doi:10.3390/plants10050964_

Round 1
Reviewer 1 Report
Dear Authors,
The paper is well written and presents the results clearly. I have some comments and suggestions:
1. The qualitiy of the figures (2,3,4) should be improved: bigger fonts and numbers, Y-axis title could help the understanding. I suggest changing the colors too. The numbers of the Y-axis are unreadable.
2. An overview figure about the different developmental stages of the examined embryos, proembryos and ESs (3-10 DAAPS, 3 DAP etc.) would be very useful for the readers.
line 90: Please, give the meaning of DAAPS abbr. in the text at the first mention.
line 206: in maize proembryos and ESs?
line 305: I think, that a short description of calculation by the 2-ΔΔCT method in the supplementary material could provide support in the clear understanding of the figures.
Author Response
I would like to thank you for helpful comments and suggestions.
Sincerely, Mikhail Chumakov

Reviewer 2 Report
Dear Authors,
You have done extensive work of importance in my opinion and I think yours obtain results and conclusions could interest many researchers and readers. There are fine observations that merit to be published in “Plants”. Although the work is interesting, I think that You should take a count small modification of this article. I recommend publishing it in "Plants" after correcting listed below suggestions:
- Because there are a lot of abbreviations in the text, I suggest to create the abbreviation index
- There is lack information about application meaning of apomixis (e.g. about maintaining heterosis) both in the Introduction both in Results and Discussion chapter.
Introduction:
Line 31: „…were very interesting as a model” –it would be good to elaborate on this thought
Results and Discussion
Line 97: Figure 1 – this figure is of bad quality and there is lack of scale - it should be corrected
Line 115: „Normally” – this expression shoud be changed
Line 120-123: it would be interesting to keep theese embryo sacs in the tissue culture conditions on the media which could allow them to develop.
Line 206: „2.3.2. Histone deacetylase (hdt104) and histone linker (hon101) expression in” - ?
Matrials and Methods
Authors should be more precise on the dates of planting. More information is needed on crop management (fertilization, irrigation, etc.)
Author Response
I would like to thank you for helpful comments and suggestions. Please see the attachment.
Sinmcerely, Mikhail Chumakov
